# Improved Flame-Retardant and Ceramifiable Properties of EVA Composites by Combination of Ammonium Polyphosphate and Aluminum Hydroxide

**DOI:** 10.3390/polym11010125

**Published:** 2019-01-12

**Authors:** Feipeng Lou, Kai Wu, Quan Wang, Zhongyu Qian, Shijuan Li, Weihong Guo

**Affiliations:** Polymer Processing Laboratory, Key Laboratory for Preparation and Application of Ultrafine Materials of Ministry of Education, School of Materials Science and Engineering, East China University of Science and Technology, Shanghai 200237, China; loufeipeng@yeah.net (F.L.); 13122320038@163.com (K.W.); wangquan_ecust@163.com (Q.W.); 18852952927@163.com (Z.Q.); shijuanliecust@163.com (S.L.)

**Keywords:** ethylene-vinyl acetate, composites, ceramifiable, flame-retardant, sintering, ceramics

## Abstract

Ceramifiable flame-retardant ethylene-vinyl acetate (EVA) copolymer composites for wire and cable sheathing materials were prepared through melt compounding with ammonium polyphosphate (APP), aluminum hydroxide (ATH) and fluorophlogopite mica as the addition agents. The effects of ammonium polyphosphate, alumina trihydrate, and APP/ATH hybrid on the flame retardant, as well as on the thermal and ceramifiable properties of EVA composites, were investigated. The results demonstrated that the composites with the ratio of APP:ATH = 1:1 displayed the best flame retardancy and the greatest char residues among the various EVA composites. The tensile strength of the composites was 6.8 MPa, and the residue strength sintered at 1000 °C reached 5.2 MPa. The effect of sintering temperature on the ceramifiable properties, microstructures, and crystalline phases of the sintered specimen was subsequently investigated through X-ray diffraction, Fourier transform infrared, and scanning electron microscopy. The XRD and FTIR results demonstrated that the crystal structure of mica was disintegrated, while magnesium orthophosphate (Mg_3_(PO_4_)_2_) was simultaneously produced at an elevated temperature, indicating that the ceramization of EVA composites had occurred. The SEM results demonstrated that a more continuous and compact microstructure was produced with the rise in the sintering temperature. This contributed to the flexural strength improvement of the ceramics.

## 1. Introduction

Ethylene vinyl acetate copolymers (EVA) are widely utilized in daily applications, such as buildings, furniture, cable jacketing, and shoes, due to the corresponding excellent electrical insulation, good processing capability, and molding properties [1,2,3,4]. However, EVA resins are easily flammable, similar to most polyolefin materials, due to the corresponding chemical compositions, which dramatically limit applications in the wire and cable industry. Consequently, it is imperative to improve the flame retardancy of EVA. Among flame-retardant additives, intumescent flame retardants (IFRs) have attracted significant research attention due to their advantages, such as low emission of toxic gas and high flame-retarding efficiency [5]. By contrast, the mechanical strength of expansion char layer is quite low and the final material cannot sustain the impact of an external force, which results in fires. A new kind of ceramifiable EVA composite can meet the demand in special fields, such as construction building, power transmission, and fireproof sheaths for cables. Ceramifiable polymer composites possess dual performance, being a polymer at room temperature and a ceramic at high temperature [6]. The ceramifiable polymer composites would transform the pyrolysis product into a stiff and porous ceramic in case of fire and the formed ceramic can isolate fire to maintain the electrical circuit integrity for up to 60 min.

The ceramifiable polymer composites are commonly composed of polymer matrix, inorganic fillers, fluxing agents, and other additives. The inorganic fillers include mica, kaolin, wollastonite, montmorillonite, and laponite platelets [7,8,9,10]. The fluxing agents (glass dust, zinc borate, and ammonium polyphosphate) start to soften and produce a high amount of liquid along with the increase in of sintering temperature [11,12]. The liquid phase joins the inorganic and other fillers together, preventing the combustible gas from escaping. Most researchers have focused on the study of the preparation and sintering mechanism of ceramifiable silicone rubber composites. Cheng et al. first prepared the ceramifiable silicone rubber composites and improved the mechanical strength of the formed ceramic through the fluxing agent addition with low softening point temperature [13,14,15]. Wang et al. systematically researched the sintering and mechanical properties of SR composites with five different inorganic fillers. Moreover, it was pointed out that the addition of a high amount of glass powder in the composites would increase the mechanical strength of the ceramic residues, but it would be accompanied with a huge shrinkage [16]. In contrast, the preparation of silicone rubber composites requires a special extrusion and vulcanization process. Therefore, the processing technology is relatively complex. Moreover, the processing cost of the silicone rubber composite is relatively high. Hence, studies on non-silicone-based composites have received increased attention in recent years due to the easy processing ability and low price. Li et al. added glass powder, montmorillonite, and mica powder to the EVA matrix to prepare a novel ceramifiable EVA composite for insulated wires. Also, the ceramifying process of sintered specimens at different temperatures was systematically studied [17]. Wang et al. prepared the ceramifiable EVA composites by adding glass frits and ammonium polyphosphate (APP) into the EVA matrix and the influence of APP on the ceramifiable properties of the composites was studied [18]. However, the addition of glass powder into the polymer would increase the electrical conductivity of the cable and cause the large linear shrinkage of the sintered specimen, which limits the practical applications. In addition, ceramifiable EVA composites exhibit excellent fire resistance at high temperatures; however, the flame retardancy of the composites at low burning temperatures was also important.

In this study, a novel ceramifiable flame-retarded EVA composite was prepared through the addition of ammonium polyphosphate, aluminum hydroxide, and fluorophlogopite mica to the EVA matrix. The effects of APP/ATH ratio on the flame retardancy, mechanical, thermal, and ceramifiable properties of the EVA composites were investigated. The ceramifying process of the EVA composites at different sintering temperatures was analyzed through mechanical testing, scanning electron microscopy (SEM), X-ray diffraction (XRD), and Fourier transform infrared spectroscopy (FT-IR).

## 2. Materials and Methods

### 2.1. Materials

Ethylene–vinyl acetate copolymer (Elvax 260) containing 28 wt % of vinyl acetate was supplied by DuPont Company (Wilmington, Delaware, USA). Ammonium polyphosphate (APP-II, polymerization degree exceeds 1500) was provided by the Hangzhou JLS Flame Retardants Chemical Co. Ltd. (Hangzhou, China). Fluorophlogopite mica was purchased from the Lingshou Jiali Processing Factory (Shijiazhuang, China). Aluminum hydroxide (ATH) was supplied by the Sinopharm Chemical Reagent Co. Ltd. (Shanghai, China). The morphology of APP, ATH, and fluorophlogopite mica are clearly shown in Figure 1 and Table 1 lists the chemical compositions of fluorophlogopite mica.

### 2.2. Sample Preparation

Before processing experiment, all the fillers were dried in the oven for 12 h at 80 °C. The neat EVA was melted in a HAAKE torque rheometer at 150 °C. Then the APP, ATH and fluorophlogopite mica fillers were added into the mixer to obtain a homogeneous mixture. The resulting mixtures were operated at 150 °C for 10 min under 10 MPa pressure. The compounding formulas of ceramifiable EVA composites are summarized in Table 2.

### 2.3. Sintering Test

Samples were cut into a rectangle shape and then sintered in a muffle furnace (Sigma instrument manufacture Co. Ltd. (Luoyang, China). The samples were heated to the target temperature at a heating rate of 10 °C min^−1^, held for 60 min, and then cooled down to ambient temperature.

### 2.4. Characterization

Chemical compositions of the fluorophlogopite mica was studied by the X-ray fluorescence method with a Shimadzu XRF-1800 spectrometer (Kyoto, Japan). The flexural strength was carried out on a Universal Testing Machine CMT4204 according to the standard of GB/T 9596-2006 and the cross-head speed was 0.5 mm min^−1^. All the results were the average of five tests. The linear shrinkage of the composites was calculated with the following equation: Linear shrinkage (%) = (L_0_ − L_1_)/L_0_ × 100%; L_0_: the length of the composites before sintering; L_1_: the length of the specimens after the sintering at different temperature. The limiting oxygen index (LOI) values were carried out on an FTT 0078 (FTT Company, East grinstead, UK) according to ISO 4589-2. The vertical test was performed on a CZF-4 type instrument (Jiangning Analysis Instrument Company, Nanjing, China) on the specimens of 130 × 13 × 3.2 mm^3^ according to the UL 94 test standard. MCC test were performed using a FAA Microscale Combustion Calorimeter (FTT Company, East grinstead, UK) and the samples about 5 mg were heated to 800 °C at a heating rate of 1 °C s^−1^ in a stream of nitrogen flowing at 80 cm^3^·min^−1^. Tensile tests were measured using a Universal Testing Machine (CMT4204, Shenzhen Sans Testing Machine Co., Ltd., Shenzhen, China) according to GB/T 1040.2-2006, and the results were repeated five times. Apparent porosity of the ceramic residue was characterized using Archimedes’ method and the water was immersion medium. Thermogravimetric analysis (TGA) was performed using an STA 449F3 apparatus (NETZSCH, Selb, Germany) from room temperature to 800 °C at 10 °C min^−1^ in a nitrogen or air atmosphere. XRD patterns of EVA composites and sintered specimens at different sintering temperature were performed by X-ray diffractometer (D/max-2550, Rigaku, Akishima, Japan) using Cu Kα radiation (λ = 1.54178 Å). The cross-section morphology of the sintered specimen and the char residues after cone calorimeter test were analyzed by field emission scanning electron microscopy (FESEM, S4800, Hitachi, Tokyo, Japan). The accelerated voltage is 15 kV and all samples were sputtered with a thin layer of gold before being examined. FTIR spectra of the residues were recorded with KBr powder by using a Nicolet 6700 spectrometer (Thermo Fisher Scientific, Waltham, MA, USA). All the samples were scanned from 4000 to 400 cm^−1^. Impact resistance tests were measured using an electronic Charpy impact tester (Suns, Shenzhen, China) according to ASTM D256-10e1 and the results were repeated five times. The cone calorimeter (Stanton Redcroft, East grinstead, UK) tests were performed according to ISO 5660 standard procedures. The samples of dimensions 100 × 100 × 3 mm^3^ was wrapped in aluminum foil and exposed horizontally to different incident heat fluxes of 35 kW·m^−2^ and 50 kW·m^−2^ in the presence of an igniter’s spark.

## 3. Results and Discussion

### 3.1. Flexural Strength and Linear Shrinkage of the Sintered Specimens

Figure 2 presents the effects of APP/ATH ratio on the linear shrinkage and flexural strength of the EVA composites sintered at 1000 °C. As presented in Figure 2, the EVA2 sample containing ATH only exhibited linear expansion, which indicated the bloating of the sintered residue. Contrary to the EVA2 sample, the EVA1 sample residue containing APP only, presented high residual shrinkage. It was noteworthy that the linear shrinkage changed from negative values to positive values as the APP filler content increased among the EVA5, EVA6, and EVA7 samples. In the case of samples EVA4, EVA5 and EVA6, the linear shrinkage was positive, indicating that a degree of sintering occurred in these samples. Furthermore, the EVA7 sample exhibited limited linear expansion. The results indicated that the linear shrinkage was inhibited through the APP addition. The phenomenon was in accordance with the research of Zhao [19]. In general, the formed ceramic at high temperature required a certain mechanical strength for structural use. The shape of the EVA1 sample after sintering at 1000 °C was irregular, which could not meet the testing requirements for the flexural strength. It could be observed that the EVA5 (APP:ATH = 1) sample presented the maximum flexural strength value (5.24 MPa) and the minimum value for the EVA2 was achieved (0.53 MPa) among all formulations. In summary, the EVA5 exhibited acceptable linear shrinkage and superior flexural strength at high temperature.

### 3.2. Mechanical Properties and Flammability of EVA Composites

The effects of APP/ATH ratio on the flame retardancy and mechanical properties of the EVA composites are illustrated in Table 3. The EVA2 sample presented the worst tensile strength and the EVA6 sample presented the best tensile strength among all formulation samples. By contrast, it could be concluded that the ratio of APP/ATH had a slight effect on the mechanical strength among the EVA composites. The tensile strength of sample EVA5 exceeded 6 MPa. It was indicated that the composites could meet the requirements of sheathing materials for wires and cables.

The flame-retardancy properties of the EVA composites were characterized by oxygen index and vertical burning ratings (UL-94). The oxygen index beyond 26 is generally considered to be a nonflammable material [20]. As presented in Table 3, the LOI value of the neat EVA was only 20.5%, whereas the LOI value of EVA2 only containing ATH was as high as 28.3%. The LOI value was slightly higher for the samples containing APP compared to the sample EVA2 only containing ATH. Moreover, the sample EVA5 with APP/ATH = 1 presented the highest LOI value of 29.7%, which could be classified as a self-extinguishing polymer. It could also be observed that the neat EVA, EVA1, and EVA2 samples did not exhibit any rating, whereas EVA6 passed the V-1 rating in the UL-94 tests. However, the EVA5 sample could pass the V-0 rating among the various formulations. According to the LOI and UL-94 tests, the prepared ceramifiable EVA5 composites presented excellent flame retardancy.

The microscale combustion calorimeter (MCC) is a convenient small-scale testing instrument to evaluate the flammability properties of polymer materials. The heat release rate (HRR) plots of pure EVA and EVA composites are shown in Figure 3 and the corresponding combustion data are presented in Table 4. It could be observed that the peak heat release rate (PHRR) and the total heat release (THR) of all EVA composites were lower than the neat EVA. The EVA2 sample with only ATH had the lowest PHRR_1_ and the EVA5 sample presented the lowest PHRR_2_ among all the EVA composites. The two peak HRR values were also present for the samples with APP. The first stage could be attributed to the earlier decomposition of the flame retardant and the second stage was assigned to the protective phosphorus–carbon layer destruction. The THR value of the neat EVA was 38.2 kJ g^−1^, while the composites with APP and ATH presented lower values of THR. Furthermore, the THR value of the EVA5 sample was 11.5 kJ g^−1^, the lowest value among all the composites.

### 3.3. Combustion Performance of Ceramifiable Flame-Retardant EVA Composites

A cone calorimeter test is used to compare the combustion behavior of the polymer composites. The smoke production rate (SPR), total release rate (THR), heat release rate (HRR), and mass loss rate (MLR) curves of neat EVA and EVA composites at a flux of 35 kW m^−2^ and 50 kW m^−2^ are shown in Figure 4 and Appendix A, and the corresponding data are summarized in Table 5 and Appendix A. The following discussion takes a flux of 35 kW m^−2^ as an example. It could be seen that the pure EVA burnt out within 649 s after being ignited and a sharp peak HRR (PHRR) of 843.8 kW m^−2^ was obtained at 197 s. The PHRR values of the EVA1, EVA2, and EVA5 composites decreased significantly compared with that of pure EVA. It should be noted that the PHRR value of EVA5 with APP and ATH was 157.7 kW m^−2^, higher than that of EVA1 only with APP (138.7 kW m^−2^) and EVA2 only with ATH (123.1 kW m^−2^). It was indicated that some antagonistic effects occurred between the APP and ATH and increased the PHRR value of EVA composites. Similar results could also be found in Castrovinci’s research [5]. They showed that the addition of ATH in the system (EVA/Mica/APP) would disrupt continuous and homogeneous surface shield over the polymer surface and decreased its effectiveness as a flame retardant.

Figure 4b shows the total heat release (THR) curves of EVA composites in the cone calorimeter test. It could be seen that the THR value (115.9 MJ m^−2^) of EVA0 was much higher than that of the other specimens. Compared with EVA0, the THR value of EVA1 with only APP was greatly reduced. Regarding the EVA2 sample, the value of THR (65.5 MJ m^−2^) was highest among all the composites and the PHRR value of EVA5 was lower than the EVA2 only with ATH, indicating that the flame spread slowed down due to the addition of APP in the system.

The smoke production rate (SPR) curves of neat EVA and EVA composites are illustrated in Figure 4c. It was noticeable that the peak of SPR (PSPR) for EVA1, EVA2, and EVA5 composites was dramatically reduced compared with neat EVA. Compared with neat EVA (7.5 × 10^−2^ m^2^ s^−1^), the PSPR values of the EVA5 and EVA2 composites were remarkably decreased to 3.3 × 10^−2^ m^2^ s^−1^ and 0.6 × 10^−2^ m^2^ s^−1^, respectively. Moreover, it was observed that the peak SPR values of EVA1 (2.1 × 10^−2^ m^2^ s^−1^) and EVA5 (3.3 × 10^−2^ m^2^ s^−1^) composites containing APP were higher than that of EVA2 only with ATH in the combustion process. The results could be explained by the fact that APP decomposed at low temperature to form some smoke particles [21].

Figure 4d presents the mass loss curves of all the samples in the CCT. It could be seen that sample EVA0 lost its mass faster than the other three samples (EVA1, EVA2, and EVA5). The char residue of the EVA1 (67.1%) only with APP improved significantly compared with that of neat EVA (8.9%). In addition, the char residue of sample EVA2 was the lowest among all the samples, which was attributed to the absence of APP acting as charring agent in the system.

The digital photographs of the char residues for the neat EVA and EVA composites after CC test at a flux of 35 kW m^−2^ and 50 kW m^−2^ are presented in Figure 5 and Appendix A, respectively. The following discussion takes flux of 35 kW m^−2^ as an example. From Figure 5a, there was no residue left for EVA0 after CC test, while a char layer was formed for EVA1 and EVA5 composite. In Figure 5c, a grayish-white residue was formed without significant charring, and big holes or cracks could also be observed. Furthermore, the expanded carbon layer generated by combustion of EVA5 was cracked and the residue of the EVA1 composite was obviously more compact than that of the EVA5 composite.

The scanning micrographs of the top surface of char residues after cone calorimeter tests at a flux of 35 kW m^−2^ and 50 kW m^−2^ are presented in Figure 6 and Appendix A, respectively. The following discussion takes a flux of 35 kW m^−2^ as an example. It was apparent that the char residue from EVA2 only with ATH displayed a loose and cracked structure. The residue of composite EVA1 showed a continuous structure with a small number of holes. Compared with the residue of EVA1, the carbonaceous residues of EVA5 with APP and ATH became looser and less compact, which indicated that the addition of ATH in the system (EVA/Mica/ATH) could not increase the compactness of the carbon layer.

Based on the above analysis, the EVA1 only with APP had better flame retardancy than EVA5. However, EVA1 composites only with APP still could not meet the fire resistance requirements for shape and dimensional stability at high temperatures. Hence, according to comprehensive analysis, the composite EVA5 showed excellent fire resistance and flame-retardant performance.

### 3.4. Thermal Analysis

The thermogravimetric analysis (TGA) curves of different inorganic fillers under N_2_ and air atmosphere are presented in Figure 7. The initial decomposition temperature (T_5%_) was defined as the temperature at 5 wt % of the mass loss and the T_max_ was the temperature at the maximum mass loss rate for the polymer. It could be seen that the thermal degradation curve of the APP, ATH, and mica under air and N_2_ conditions were similar. The thermal degradation of APP included two steps: the first stage was attributed to the release of ammonia and water from APP, whereas the second stage was assigned to the decomposition of the polyphosphoric acid chain with the maximal mass loss rates at 637 °C [21,22]. The initial decomposition temperature of ATH was 275 °C, with two maximal mass loss rates at 305 °C and 520 °C, and the final residue was 65.9% under nitrogen conditions [23]. However, the initial decomposition temperature of ATH in air was 251 °C, which was much lower than in the nitrogen atmosphere. The mica sustained a one-step mass loss, which was assigned to the dehydroxylation of mica, causing lattice damage and vacancies [24].

The TGA curves of the neat EVA and EVA composites under N_2_ and air atmosphere are presented in Figure 8; the corresponding data are given in Table 6. It could be observed that the thermal degradation of neat EVA mainly included two stages in air and N_2_ atmosphere. The first stage was caused by the loss of acetic acid, due to the decomposition of vinyl acetate groups [25]. The second step was assigned to the degradation of polyethylene chains, leading to the complete polymer volatilization. The maximum weight loss temperatures (T_max_) for the two decomposition steps were 350 °C and 468 °C, respectively, and almost no residue was left. Compared with the initial decomposition temperature (T_5%_) in air, the value of T_5%_ was greatly reduced, which was attributed to the oxidation of the composite in air. The following discussion takes EVA composites under N_2_ atmosphere as an example. In the case of sample EVA1 containing APP only, the initial decomposition temperature was 330 °C, slightly lower than that of neat EVA (335 °C). This phenomenon could be explained by the APP water release under thermal degradation and the hot water hydrolysis that led to the earlier degradation of the EVA [26]. In addition, the second reason was that the phosphorus groups in the composite decompose at relatively low temperatures to generate heat-resistant char [27]. Compared to the neat EVA, the sample with APP only (EVA1) had higher thermal stability beyond approximately 357 °C and a significantly higher amount of residue at 800 °C. The thermal degradation of sample EVA2 had three steps, but EVA1 with only APP showed two decomposition stages. The three thermal degradation steps were attributed to water release, EVA matrix degradation, and ATH decomposition, respectively. It should be noted that the ratio of APP/ATH highly affected the thermal behavior of the EVA composites. As an example, the initial degradation temperature decreased as the ATH content increased in the composites. Moreover, the residue of sample EVA2 with ATH only was quite a bit lower than the other EVA composites containing APP. The EVA5 sample presented the highest residue (58.1%) under N_2_ atmosphere among all the EVA composites.

The calculated and experimental TG curves of the EVA5 sample in N_2_ and air are presented in Figure 9. The curve of the EVA composites in the N_2_ atmosphere was similar to the curve of the EVA composites in air in the first stage. The following discussion takes EVA composites under N_2_ atmosphere as an example. The maximum weight loss rates of the experimental TG curve in the first stage were higher than the calculated mass loss rate. Also, the initial decomposition temperature of the experimental TG curve was lower than the calculated curve. This was attributed to certain antagonistic effects occurring among the fillers or the degradation product of the EVA composites. The calculated and experimental curves were nearly identical at the second degradation stage, indicating that no chemical reactions occurred between the APP and ATH below 537 °C. It was interesting that the third degradation stage of the experimental curve disappeared, whereas it still existed in the calculated TG curve. This suggested that the pyrolysis products of APP were already involved in the system interaction, leading to an apparent reduction of the mass loss rate of the EVA5 composites as well as to the higher char residue at 800 °C.

According to the comprehensive analysis, the EVA5 sample presented excellent ceramifiable properties, flame retardancy, and thermal properties. Hence, the optimal formulation (EVA5 sample) was selected to further explore the ceramifying process of the EVA composites at different temperatures, as presented in the next sections.

### 3.5. Surface Morphology of the Sintered Specimen

Figure 10 presents the surface morphologies of the EVA5 sample and ceramic residues sintered at different temperatures. It could be observed that the EVA5 composite before sintering was white and smooth. The surface color of the sintered specimen changed from white to black and the surface morphology became irregular; there was a slight expansion with the increasing sintering temperature from room temperature to 400 °C. When the sintering temperature was increased to 500 °C, the color of the sintered specimen became gray, which was attributed to the carbon residue layer covering the surface of the polymer continuing to burn under in. The shape of the sintered sample did not change greatly, and the sintered body became grayish as the temperature increased from 500 °C to 600 °C. As the sintering temperature increased to 700 °C, the color of the sintered specimen changed from gray to white, and the surface was smooth and could be maintained at a high temperature of 1000 °C. In conclusion, the shape of the ceramic residue was similar to the composite EVA5 at room temperature, which indicated that it maintained good shape stability at different sintering temperatures. The reason was that mica could provide a self-supporting skeleton structure and retain the original shape of the materials after the EVA had decomposed.

### 3.6. Linear Shrinkage

The linear shrinkage of the EVA composites at various temperatures from 400 °C to 1000 °C is presented in Figure 11. The linear shrinkage value of the residue sintered at 400 °C was positive, indicating that the sintered specimen expanded due to the thermal degradation of the EVA matrix and the fillers. As the sintering temperature increased to 700 °C, the linear expansion of the sintered specimen increased and reached the maximum value. Compared to the results for the lower temperature, the linear shrinkage apparently changed from negative to positive values as the temperature increased from 900 °C to 1000 °C and the value of linear shrinkage was 1.77%. This demonstrated that a certain amount of sintering had occurred in the ablation process, causing obvious shrinkage. Hence, it could be concluded that the EVA5 sample maintained its excellent dimensional stability at various sintering temperatures.

### 3.7. Apparent Porosity and Mechanical Strength of Sintered Sample

The apparent porosity, flexural strength, and impact strength of the sintered specimens are presented in Figure 12 and Figure 13. It should be noted that the apparent porosity of the sintered specimen slightly decreased with the increasing sintering temperature from 400 °C to 500 °C, which was attributed to the greater degradation of filler APP, ATH, and EVA matrix at a high temperature and resulted in a higher hole concentration. However, it could be observed that the apparent porosity decreased from 60.3% to 52.2% as the sintering temperature increased from 600 °C to 900 °C. The sintering temperature continued to increase to 1000 °C; the apparent porosity of the sintered specimens decreased abruptly from 52.2 to 46.4%. The flexural strength of the sintered specimens increased from 1.75 to 5.24 MPa with the increasing temperature. The flexural strength value reached the maximum when the sintering temperature reached 1000 °C. As shown in Figure 13, the impact strength of the ceramic residue increased from 0.9 J·m^−1^ to 6.4 J·m^−1^ as the sintering temperature increased from 400 °C to 1000 °C. As a consequence, the prepared porous ceramics with a relatively high mechanical strength and a certain degree of apparent porosity were expected to provide excellent flame retardancy and thermal insulation.

### 3.8. XRD Analysis

The crystal pattern of the sample EVA5 composites at room temperature and the sintered specimens are shown in Figure 14. In case of EVA1 sample, the fluorophlogopite mica and magnesium‒ammonium polyphosphates (NH_4_Mg(PO_3_)_3_) crystals were presented as major and minor phases at 400 °C, respectively [28]. The residue components did not change when the sintering temperature was increased to 600 °C. When the sintering temperature increased from 600 °C to 800 °C, a diffraction peak corresponding to magnesium pyrophosphate (Mg_2_P_2_O_7_) was observed; meanwhile, the diffraction peak representing NH_4_Mg(PO_3_)_3_ disappeared. This indicated that the chemical reaction between NH_4_Mg(PO_3_)_3_ and mica had happened and there was a loss of ammonia [29]. Furthermore, the diffraction peak of aluminum phosphate (AlPO_4_) was also observed, which was due to the reaction between active mica and decomposition products of APP. Moreover, the diffraction peak intensity of magnesium pyrophosphate (Mg_2_P_2_O_7_) and AlPO_4_ became obvious and the peak of mica completely disappeared at 1000 °C. Hence, the Mg_2_P_2_O_7_ and AlPO_4_ crystals became the main phases at high temperature in the sintered ceramic-like body.

In EVA2 sample, the main products in the residue were boehmite (AlOOH) and mica at 400 °C. At this temperature, the appearance of AlOOH was ascribed to the degradation of aluminum hydroxide (ATH). When the sintering temperature was increased from 600 °C to 800 °C, the characteristic diffraction peaks of alumina (Al_2_O_3_) were observed due to complete decomposition of ATH. Further heating to 1000 °C resulted in the formation of magnesium silicate (Mg_2_SiO_4_), but the characteristic diffraction peaks of mica did not change. This implies that the degradation products of ATH did not react with mica and could not be linked together to form a coherent ceramic structure, leading to the low mechanical strength of the residue.

As presented in Figure 10c, all characteristic peaks of the ceramifiable EVA5 composites were consistent with the raw phase compositions of the fluorophlogopite mica, the ATH, and the APP. When the sintering temperature was increased from room temperature to 400 °C, the diffraction peaks of APP and ATH disappeared, whereas the characteristic peak of AlOOH was observed at the same time, which could be attributed to the thermal degradation of APP and ATH. Moreover, it should be noted that the diffraction peaks of NH_4_Mg(PO_3_)_3_ appeared at 400 °C. When the sintering temperature was further increased to 500 °C, the phase composition of the sintered specimen did not change. However, the diffraction peak of AlPO_4_ appeared when sintering was carried out at 600 °C. The AlPO_4_ crystal formed due to the chemical reaction between the pyrolysis products of APP and AlOOH. The results were in agreement with the TG results. Further crystal transformations were observed in the mixture at 700 °C, accompanied by the NH_4_Mg(PO_3_)_3_ disappearance and the appearance of magnesium pyrophosphate (Mg_2_P_2_O_7_). Upon further heating to 800 °C, the composition phase of the sintered specimens did not change. As the temperature was further increased to 900 °C, the intensities of mica and Mg_2_P_2_O_7_ were apparently reduced, while a low amount of magnesium orthophosphate (Mg_3_(PO_4_)_2_) was simultaneously formed. In addition, an amorphous hump was apparently diffused in the sintered specimen, which indicated that the liquid phase was produced in the residues, which could play the role of mica and the inorganic filler connection as well as densify the sintered residue. When the sintering temperature continued to increase to 1000 °C, new peaks at 2θ = 20.3°, 20.6°, 24.3°, and 25.8°, belonging to the Mg_3_(PO_4_)_2_, were clearly detected and the intensity of mica was further reduced [30,31]. The disintegration of fluorophlogopite mica and the simultaneous appearance of AlPO_4_ and Mg_3_(PO_4_)_2_ phases indicated the ceramization process of EVA composites in the sintered specimen, which resulted in the formation of a complete ceramic phase at high temperature.

### 3.9. FTIR Analysis

The FTIR spectra of the specimens sintered at different temperatures from 400 °C to 1000 °C are presented in Figure 15. As shown in Figure 15, the broader band at 3433 cm^−1^ was referred to the stretching vibration of the hydroxyl group in the sintered specimens, while the band at 1632 cm^−1^ was assigned to the H–O–H bending vibrations. The absorption bands at 1022 cm^−1^ and 479 cm^−1^ were attributed to the Si–O–Si asymmetric stretching vibration or to the bending vibration modes of the tetrahedral [SiO_4_] unit in the mica, respectively [32,33]. The FTIR spectrum of EVA1 sample exhibited that the characteristic bands at 1285 cm^−1^ and 880 cm^−1^, corresponding to the P=O and P–O–P antisymmetric vibration, disappeared when the sintering temperature was increased from 400 °C to 600 °C. In addition, the peak at 982 cm^−1^, representing P–C–O vibrations, appeared at 700 °C. These results suggest that the phosphate group reacted with other compounds and led to the formation of phosphorus-rich cross-linked structures during the decomposition process [34,35]. In addition, it should be noted that the FTIR band at 1022 cm^−1^, ascribed to mica, disappeared after sintering at 700 °C, which confirmed the degradation of mica and reaction with by-products of the APP during high-temperature sintering. In summary, the FTIR results were in good agreement with the XRD observations.

In the case of the EVA2 sample, the band at 3299 cm^−1^ and 3083 cm^−1^ were attributed to the stretching vibration peak of Al–OH after sintering at 400 °C, which suggested the formation of boehmite (AlOOH) due to the aluminum hydroxide degradation. The absorption peaks at 1022 cm^−1^ and 479 cm^−1^ did not change with increasing sintering temperature from 400 °C to 1000 °C, which indicated that the degradation products of ATH did not take part in a chemical reaction with mica. Hence, the sintered residue exhibited weak mechanical strength.

The IR spectra of EVA1 and EVA5 residues were similar. In the case of EVA5 sample, one should note that the peaks at 479 cm^−1^ and 1022 cm^−1^, corresponding to the presence of mica, were apparently reduced with increased sintering temperature. The disappearance of characteristic mica peaks indicates that the structure of fluorophlogopite mica has been destroyed and a chemical reaction had occurred with increasing sintering temperature.

### 3.10. SEM

The cross section morphologies of the sintered specimens at different temperatures are presented in Figure 16. As presented in Figure 16a, the lamellar structure of mica could be clearly observed and the fillers were randomly dispersed within the sintered specimen. It was indicated that a poor interface adhesion existed among mica and the other fillers in the sintered specimen. Compared to Figure 16a, no apparent change existed and the gaps between the fillers became higher as the sintering temperature increased to 600 °C. When the sintering temperature increased from 600 °C to 800 °C, the interfacial bonding among the mica and the other fillers was improved. As presented in Figure 16d, a large flowing liquid phase appeared around the lamellar structure filler, which could fill the hole or gaps, leading to a relatively compact and continuous microstructure. This contributed to the significant mechanical strength improvement of the sintered ceramic. To further investigate the chemical components in the sintered specimens, EDS measurements was performed. As shown in Figure 17, Mg, Al, P, O, and Si elements were distributed in the sintered body, which proved the existence of phosphorus and magnesium elements in the sintered body, which were consistent with the XRD analysis results.

## 4. Conclusions

A novel ceramifiable flame-retardant ethylene-vinyl acetate (EVA) composite with good flame retardancy and superior ceramifiable properties was successfully prepared through a simple melt-mixing method. The effects of the APP/ATH ratio on the ceramifiable properties, flame retardancy, and thermal properties of the composites were systematically investigated. The results demonstrated that the optimum ratio of APP/ATH for the EVA composites was 1:1 and the corresponding LOI value reached 29.7%. The tensile strength of the composite was 6.8 MPa and its flexural strength of the ceramic sintered at 1000 °C reached 5.2 MPa. In addition, the EVA5 composites demonstrated the lowest heat release rate and the most significant char residue among all the composites. The flexural strength of the sintered specimen apparently increased and the apparent porosity was reduced as the sintering temperature increased from 600 °C to 1000 °C. The XRD and FTIR results demonstrated that the crystalline phase of fluorophlogopite mica was destroyed and new peaks of AlPO_4_ and Mg_3_(PO_4_)_2_ crystals were formed at high temperature. The SEM results also demonstrated that the hole amount decreased as the temperature increased, leading to the bending strength improvement of the sintered specimens.

## Figures and Tables

**Figure 1 polymers-11-00125-f001:**
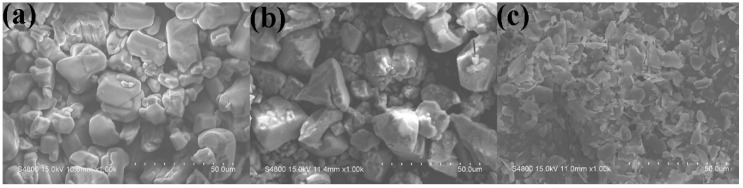
FESEM images of different inorganic fillers: (**a**) APP, (**b**) Al(OH)_3_, (**c**) mica.

**Figure 2 polymers-11-00125-f002:**
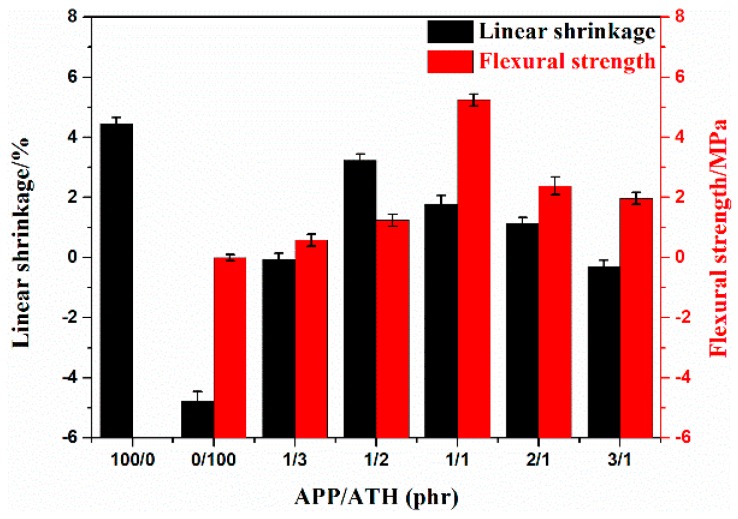
The flexural strength and linear shrinkage of the sintered specimens.

**Figure 3 polymers-11-00125-f003:**
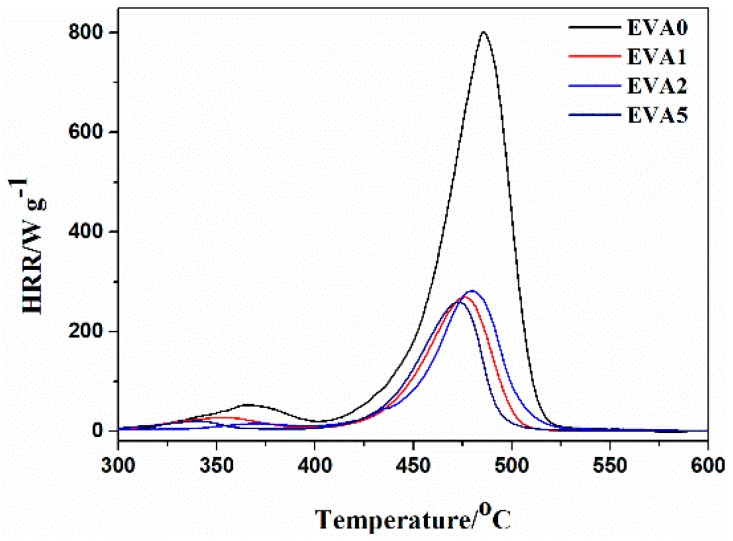
MCC curves of neat EVA and the EVA composites.

**Figure 4 polymers-11-00125-f004:**
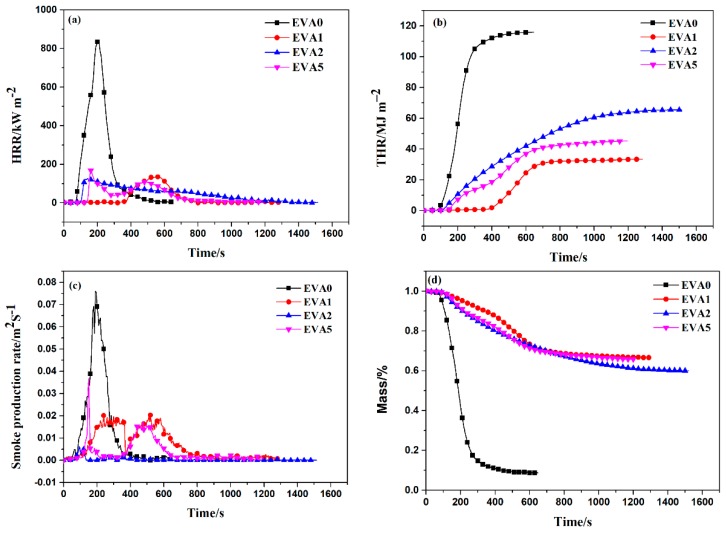
HRR (**a**), THR (**b**), SPR (**c**), and ML (**d**) curves of neat EVA and EVA composites at a flux of 35 kW m^−2^.

**Figure 5 polymers-11-00125-f005:**
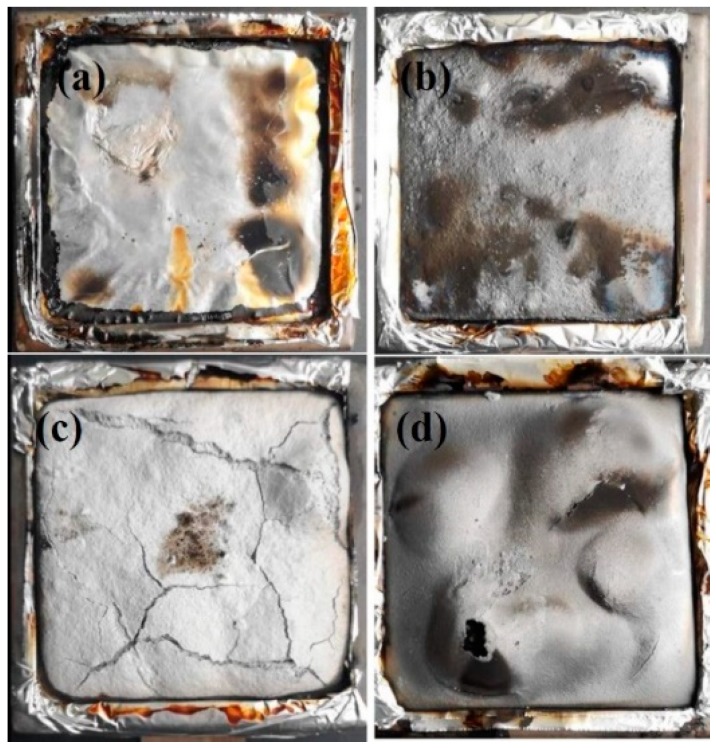
The digital photographs of the residues at a flux of 35 kW m^−2^. (**a**) EVA0; (**b**) EVA1; (**c**) EVA2; (**d**) EVA5.

**Figure 6 polymers-11-00125-f006:**
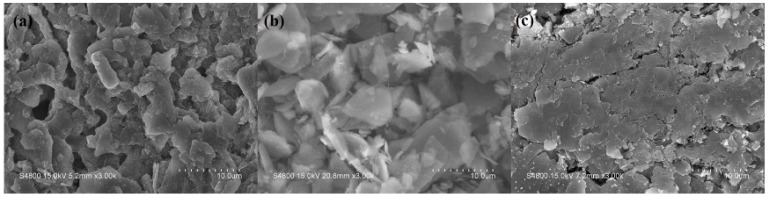
Surface morphologies of the residues after cone calorimetry at a flux of 35 kW m^−2^. (**a**) EVA1; (**b**) EVA2; (**c**) EVA5.

**Figure 7 polymers-11-00125-f007:**
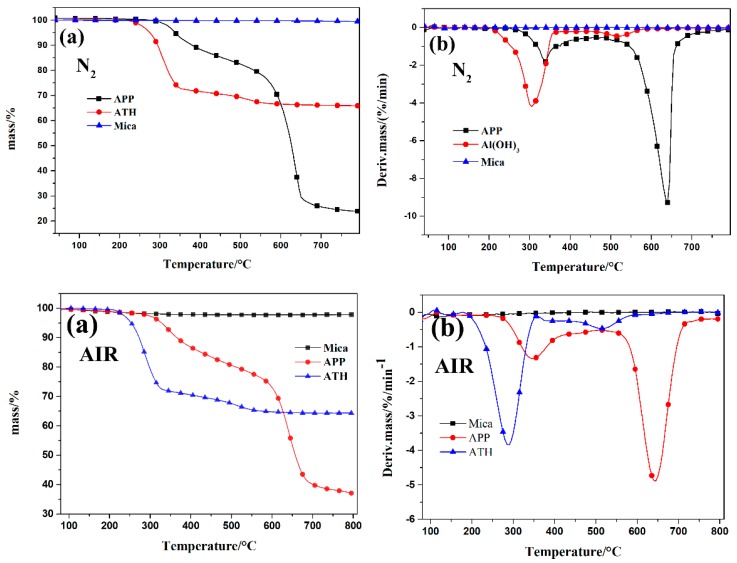
TG and DTG curves of Al(OH)_3_, APP, and mica under N_2_ and air atmosphere.

**Figure 8 polymers-11-00125-f008:**
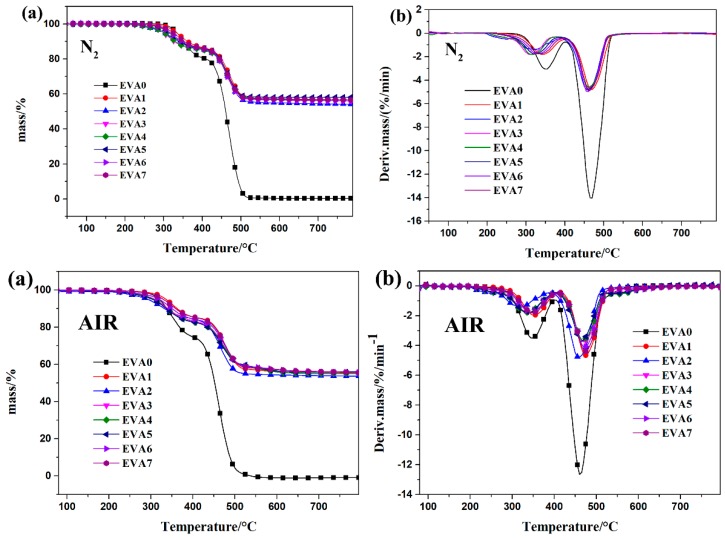
Thermal stability of neat EVA and the EVA composites under N_2_ and air atmosphere.

**Figure 9 polymers-11-00125-f009:**
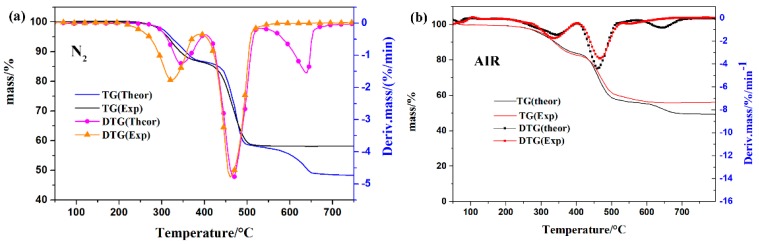
Experimental and calculated TG and DTG curves of ceramifiable EVA5 composites under nitrogen and air atmosphere.

**Figure 10 polymers-11-00125-f010:**
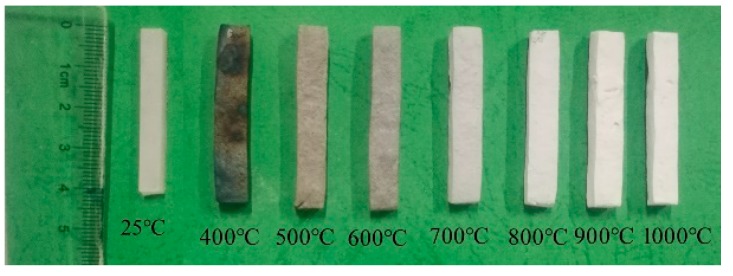
The appearance of sintered samples at different sintering temperatures.

**Figure 11 polymers-11-00125-f011:**
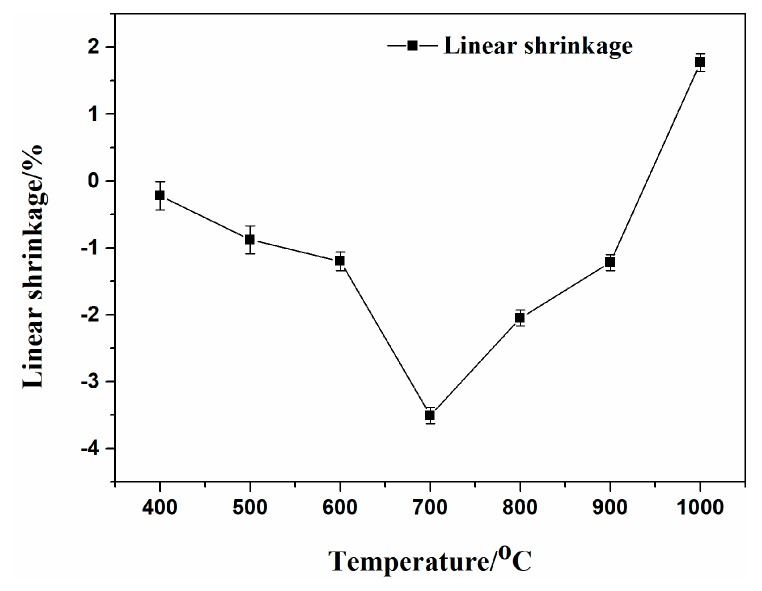
The linear shrinkage of the EVA composites at different sintering temperature.

**Figure 12 polymers-11-00125-f012:**
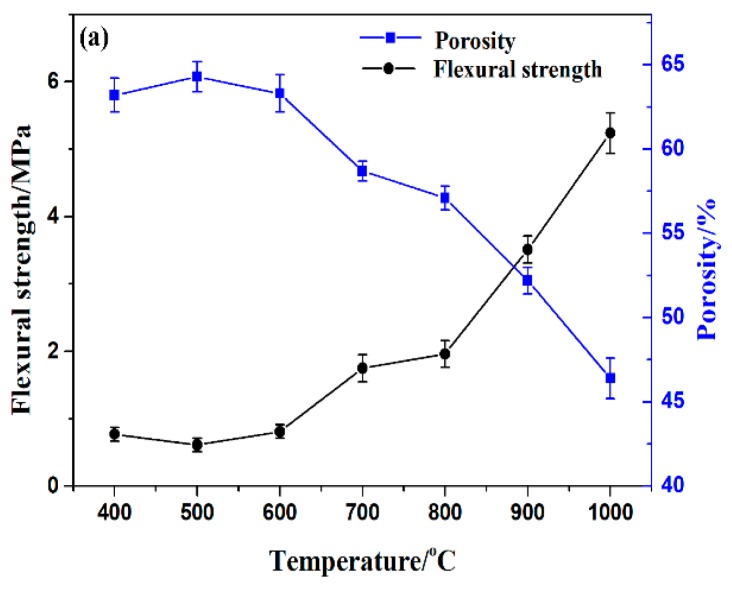
Porosity and flexural strength of the sintered specimens sintered at various temperatures.

**Figure 13 polymers-11-00125-f013:**
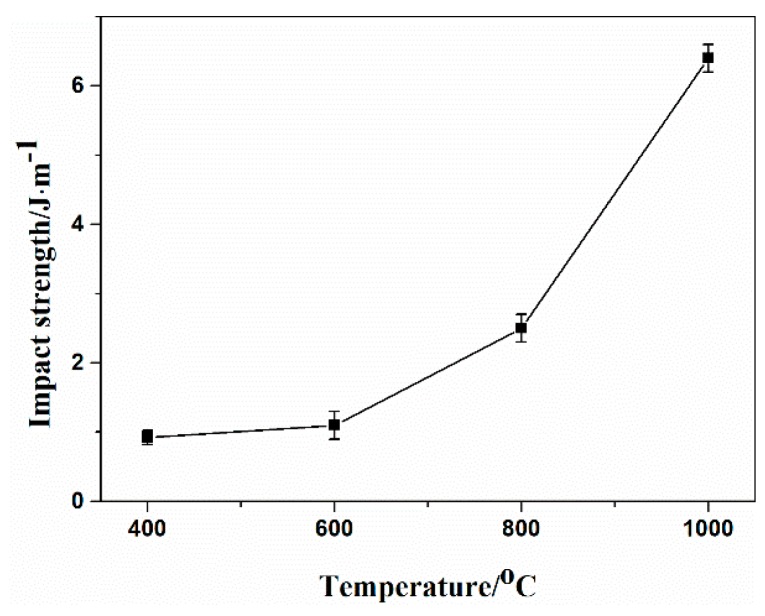
Impact strength of the sintered specimens sintered at various temperatures.

**Figure 14 polymers-11-00125-f014:**
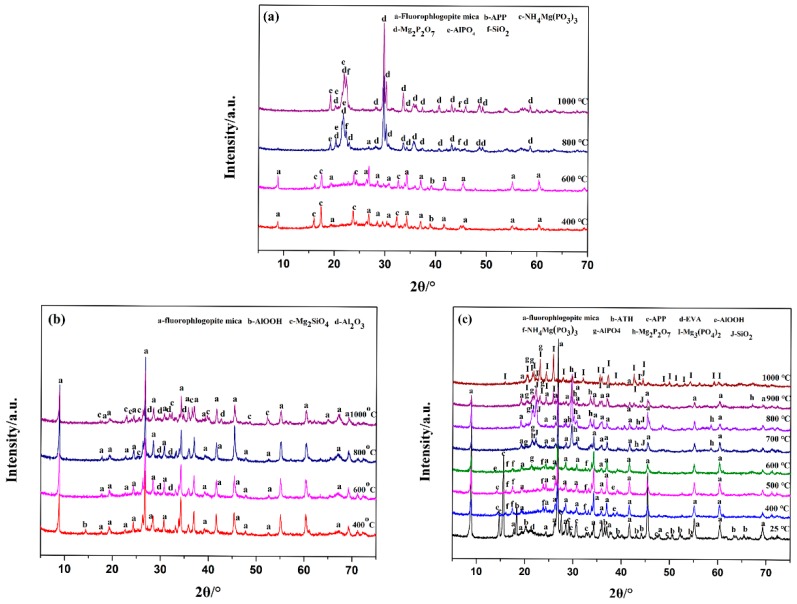
XRD spectra of sample EVA1 (**a**), sample EVA2 (**b**) and sample EVA5 (**c**) after sintered at different temperature.

**Figure 15 polymers-11-00125-f015:**
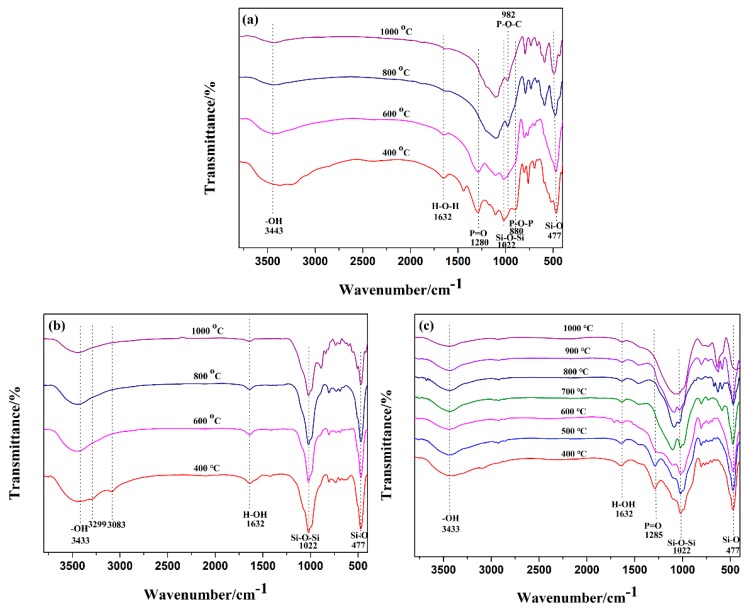
FTIR spectra of sample EVA1 (**a**), sample EVA2 (**b**), and sample EVA5 (**c**) sintered at different temperatures.

**Figure 16 polymers-11-00125-f016:**
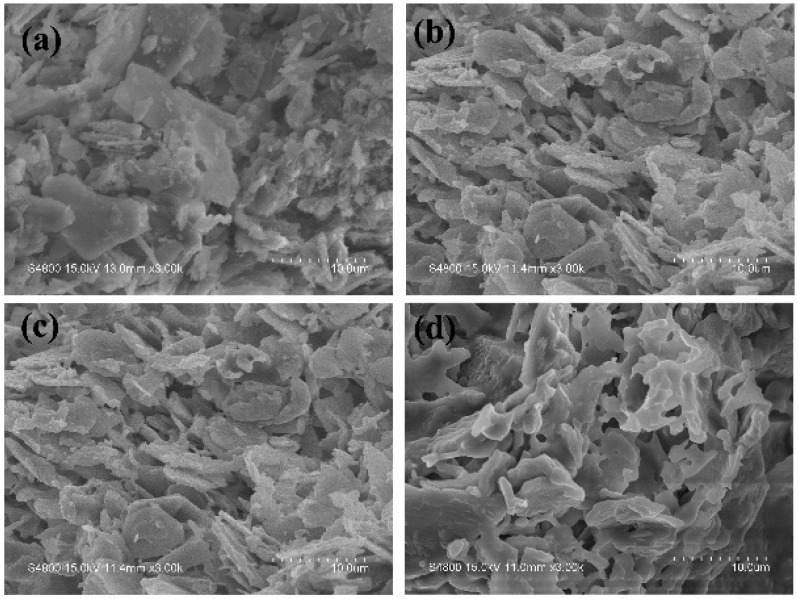
FESEM images of the sintered specimens sintered at different temperatures: (**a**) 400 °C, (**b**) 600 °C, (**c**) 800 °C, (**d**) 1000 °C.

**Figure 17 polymers-11-00125-f017:**
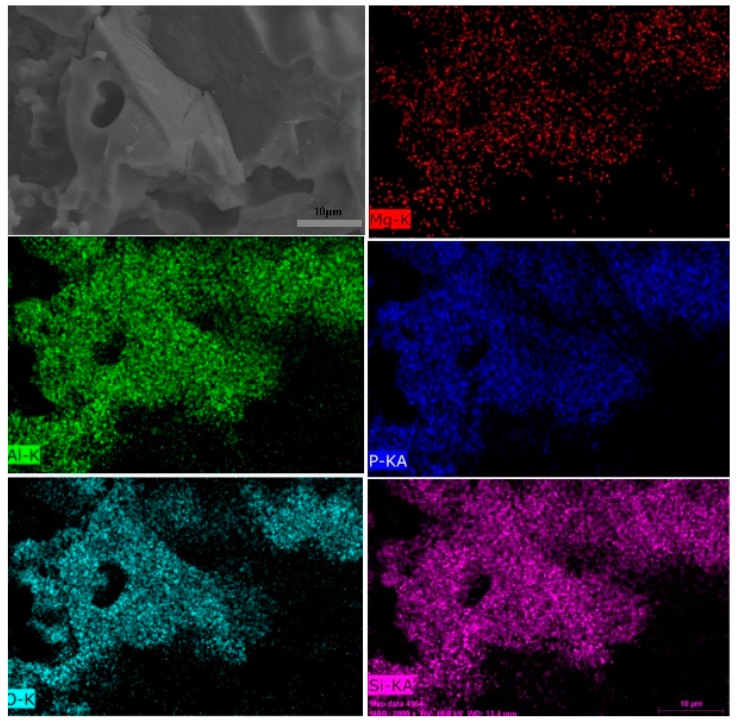
SEM–EDX results of the sintered specimens formed at 1000 °C.

**Table 1 polymers-11-00125-t001:** Chemical composition of fluorophlogopite mica.

Oxides	SiO_2_	MgO	K_2_O	Al_2_O_3_	F	CaO	Others
Contents/%	31.5	30.2	13.2	11.3	13.3	0.28	0.22

**Table 2 polymers-11-00125-t002:** Formulations of ceramifiable EVA composites.^a^

Compositions/phr	EVA/phr	Mica/phr	APP/phr	ATH/phr
EVA0	100	0	0	0
EVA1	100	100	100	0
EVA2	100	100	-	100
EVA3	100	100	25	75
EVA4	100	100	33	67
EVA5	100	100	50	50
EVA6	100	100	67	33
EVA7	100	100	75	25

^a^ Parts per hundred of EVA.

**Table 3 polymers-11-00125-t003:** Mechanical and flame-retardant properties of the EVA composites.

Samples	LOI/%	UL-94 Rating	Tensile Strength/MPa
EVA0	20.5	NC	13.6 ± 0.6
EVA1	28.3	NC	6.1 ± 0.7
EVA2	27.5	NC	5.0 ± 0.8
EVA3	28.1	V-2	5.2 ± 0.8
EVA4	28.1	V-1	5.4 ± 0.8
EVA5	29.7	V-0	6.9 ± 0.9
EVA6	29.1	V-1	7.0 ± 0.5
EVA7	28.5	V-2	6.9 ± 0.5

**Table 4 polymers-11-00125-t004:** MCC data for all samples.

Samples	PHRR_1_ (W g^−1^)	PHRR_2_ (W g^−1^)	THR (kJ g^−1^)
EVA0	51.9 ± 1.7	801.8 ± 2.1	38.2 ± 0.5
EVA1	27.5 ± 2.5	270.1 ± 2.3	13.1 ± 0.3
EVA2	15.6 ± 3	281.4 ± 3.2	13.6 ± 0.5
EVA5	20.1 ± 2	258.1 ± 3.4	11.5 ± 0.5

^a^ PHRR_1_, first peak heat release rate; PHRR_2_, second peak heat release rate; THR, total heat release.

**Table 5 polymers-11-00125-t005:** CC data of EVA and EVA composites at a flux of 35 kW m^−2^.

Samples	PHRR (kW m^−2^)	THR (MJ m^−2^)	TTI (s)	Residue (wt %)	Peak SPR (1 ×10^−2^ m^2^ s^−1^)
EVA0	843.8	115.9	71	8.9	7.5
EVA1	138.7	33.4	360	67.1	2.1
EVA2	123.1	65.5	100	59.5	0.6
EVA5	157.7	56.6	143	67.1	3.3

**Table 6 polymers-11-00125-t006:** TG data of ceramifiable EVA composites under nitrogen and air atmosphere.

Samples	T_5%_ /°C	T_max1_/°C	T_max2_/°C	Residue at 800 °C/%
**Nitrogen**				
EVA0	335	350	468	0.3
EVA1	330	345	470	55.9
EVA2	294	325	460	54.1
EVA3	298	312	460	56.1
EVA4	316	315	460	56.7
EVA5	316	325	460	58.1
EVA6	318	330	460	55.9
EVA7	327	335	460	56.6
Calculate	320	345	468	48.2
**Air**				
EVA0	317	350	460	0.1
EVA1	336	345	475	55.2
EVA2	293	325	456	52.9
EVA3	302	340	470	55.6
EVA4	304	335	471	54.9
EVA5	306	345	464	56.0
EVA6	321	345	471	55.6
EVA7	329	350	472	55.6
Calculate	295	344	460	49.2

T_5%_: the initial decomposition temperature corresponding to 5% mass loss; T_max_: the temperature at the maximum mass loss rate.

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
