# Peer review of "Improved Flame-Retardant and Ceramifiable Properties of EVA Composites by Combination of Ammonium Polyphosphate and Aluminum Hydroxide"

_polymers, 2019, doi:10.3390/polym11010125_

Round 1

Reviewer 1 Report

The paper from Lou et al. reports on the use of APP, ATH and fluorophlogopite mica as a flame retardant combination for EVA copolymers. In particular, the effects of APP/ATH ratios on the flame retradant, mechanical, thermal properties of EVA is investigated, using different analytical techniques. The manuscript shows some novelty, but the conclusions are only partially supported by the experimental data. Therefore, major revision is suggested. In particular:

- it could be useful to perform TG analyses also in air and to discuss the obtained results

- it could be also useful to perform forced-combustion (i.e. cone calorimetry) tests, starting with a heat flux of 35 kW/m2

- Table 3: too many decimal digits for indicating the tensile strength

- Table 4: please, replace "KJ g-1" with "kJ g-1"

- it could be interesting to make some impact resistance tests on the different systems

- finally, it could be very useful to perform SEM-EDX analyses on the residues after flammability (or cone calorimetry) tests.

Author Response

Reviewer 1

Dear professor:

Thank you very much for your reasonable comments. We have revised the manuscript according to your comments and suggestions. The details of the changes are shown as follows:

1.     It could be useful to perform TG analyses also in air and to discuss the obtained results.

   Response: Thanks for the referee’s useful comment. We are very sorry for the          missing the thermal stability of fillers and EVA composites in air in our        manuscript and we have added the results in the “TG analysisin Line 253, Page            8, which is marked in blue.

2.     It could be also useful to perform forced-combustion (i.e. cone calorimetry) tests, starting with a heat flux of 35 kW/m2

Response: Thanks for the referee’s meaningful advices. Cone calorimetry test is very necessary to characterize the flammability of the EVA composites in this article and we have added the related results in our manuscript, which is marked in blue in the discussion in the part “3.3 Combustion performance of ceramifiable flame-retardant EVA composites” in Line 188, Page 6.

3.     Table 3: too many decimal digits for indicating the tensile strength.

Response: Thanks for the referee’s kind suggestion. We have revised the results of the tensile strength, which are marked in blue in Table 3 in Line 172, Page 5.

4.     Table 4: please, replace "KJ g-1" with "kJ g-1"

Response: Thanks for the referee’s useful comment. We are very sorry for our mistake and have carefully checked the manuscript again to revise the mistakes thoroughly, which is marked in blue in Table 4 in Line 186, Page 6.

5.     It could be interesting to make some impact resistance tests on the different systems

Response: Thanks for the referee’s suggestion. It is necessary to test the impact resistance of the ceramic residues at different sintering temperature and the related results were added in the Fig. 13, which is marked in blue in Line 351, Page 13. The impact resistance tests were measured using an electronic charpy impact tester (Suns, China) according to ASTM D256-10e1 and the results were repeated five times [1-3]. As shown in the Fig.1, the impact strength of the ceramic residue increased from 0.9 J·m-1 to 6.4 J·m-1 as the sintering temperature increased from 400 oC to 1000 oC.

Fig. 1. Effect of sintering temperature on the impact strength of the ceramic residue.

[1] Jianhua Guo. Effect of the sintering temperature on the microstructure, properties and formation mechanism of ceramic materials obtained from polysiloxane elastomer-based ceramizable composite[J]. Journal of Alloys and Compounds 678 (2016) 499-505.

[2] Jianhua Guo. Effect of glass frit with low softening temperature on the properties, microstructure and formation mechanism of polysiloxane elastomer-based ceramizable composites[J]. Polymer Degradation and Stability 136 (2017) 71-79.

[3] Jianhua Guo. Improving the Mechanical and Electrical Properties of Ceramizable Silicone Rubber/Halloysite Composites and Their Ceramic Residues by Incorporation of Different Borates[J]. Polymers 2018, 10, 388.

6.     Finally, it could be very useful to perform SEM-EDX analyses on the residues after flammability (or cone calorimetry) tests.

Response: We are very grateful to this meaningful advice for the manuscript. According to the suggestion, we used the electron microscope instruments S4800 to analyses the residue after cone calorimetry and the results were shown in part “3.3 Combustion performance of ceramifiable flame-retardant EVA composites”, which is marked in blue in Line 246, Page 8.

   At last, thank you again for your precious time and efforts for the publication of this paper.

Reviewer 2

Dear professor:

Thank you very much for your reasonable comments.

Reviewer 3

Dear professor:

Thank you very much for your reasonable comments and criticisms. We are grateful to your positive evaluation and recommendation of this manuscript, we have revised the manuscript according to your comments and suggestions. The details of the changes are shown as follows:

1.     The submitted paper is focused on a topic that has been widely proposed in the scientific literature. The combination of ammonium polyphosphate and aluminium hydroxide has been already proven to be an efficient flame retardant for many polymers and in particular for EVA copolymers. The novelty of the paper is the use of fluorophlogopite mica, but its specific role is not sufficiently highlighted and discussed in detail.

Response: Thanks for the referee’s reasonable comment. We apologize for not  explaining the role of fluorophlogopite mica in the paper and we have added in “3.5 Surface Morphology of the Sintered Specimen and marked in red in Line 334, Page 12. In this paper, the mica could provide self-supporting skeleton structures and retain the original shape of the materials after the EVA had decomposed. As shown in the Fig. 1 in below, the prepared EVA composite could maintain good shape stability at different sintering temperatures.

Fig. 1. The appearance of sintered samples at different sintering temperatures.

2.     The term “ceramifiable” is not commonly used in the field of polymer flame retardancy. May be the Authors wanted to refer to the “charring effect” of the proposed flame-retardants. This aspect should be clarified.

Response: Thanks for the referee’s useful comment. It really confused the readers if we do not clearly describe the term “ceramifiable”. The term “ceramifiable” was different from the term “charring effect”. Ceramifiable polymer composites have the dual performance characteristics of polymers at room temperature and ceramic residue at elevated temperatures. The ceramifiable polymer composites would transform into the ceramic under fire or at high temperature. As shown the paper in Fig. 2, the EVA composite was first transformed into the carbon residue at low temperature. And with the sintering temperature increased, the carbon residue would transform into hard ceramic and possess the high flexural strength (5.24 MPa). In addition, the term “ceramifiable” is common in passive fire prevention field, the similar expressed literatures are shown in below [1-6].

Fig. 2. The appearance of sintered samples at different sintering temperatures.

[1] Ying-Ming Li. Improving fire retardancy of ceramifiable polyolefin system via a hybrid of zinc borate@melamine cyanurate[J]. Polymer Degradation and Stability 153 (2018) 325-332

[2] Dong Zhao. Ceramifiable EVA/APP/SGF composites for improved ceramifiable properties. Polymer Degradation and Stability 150 (2018) 140-147

[3] Gong Xinhao. Improved ceramifiable properties of EVA composites with whitened and capsulized red phosphorus (WCRP). RSC Adv., 2016, 6, 96984–96989.

[4] Ying-Ming Li. A novel high-temperature-resistant polymeric material for cables and insulated wires via the ceramization of mica-based ceramifiable EVA composites[J]. Composites Science and Technology 132 (2016) 116-122

[5] Rafał Anyszka. Thermal Stability and Flammability of Styrene-Butadiene Rubber-Based (SBR) Ceramifiable Composites[J]. Materials 2016, 9, 604

[6] Jinhe Wang. Mechanical and ceramifiable properties of silicone rubber filled with different inorganic fillers[J]. Polymer Degradation and Stability.2015(121) 149-156.

3.     As far as the sample preparation is concerned, two aspects should be clarified: i) the residual water content has been missed for the fillers after drying, but is an important information for further compounding; ii) the range of sample compositions should be completed. For instance, EVA1 contains 100 phr of EVA and 100 phr of both mica and APP, and EVA2 contains 100 phr of EVA and 100 phr of both mica and ATH. What is the behaviour of the sample containing only EVA and mica? Or only EVA and ATH, or EVA and APP? Testing these formulations would allow understanding the bearing of each component on fire retardancy and/or the occurrence, if any, of synergistic effects among them.

Response: We are very grateful to this meaningful suggestion for the manuscript. We quite agree with the reviewer’s viewpoint that residual water content is important for further compounding. In order to research the residual water content after drying and we recorded the weight of fillers after drying at 80 oC every 2 hours. The data of the fillers are shown in below Table 1.

Table 1. The weight of filler dried in the oven at 80 oC for different hours

Times

APP

ATH

Mica

0h

300g

300g

300g

2h

299g

299g

297g

4h

297g

298g

297g

6h

297g

298g

297g

8h

297g

298g

297g

10h

297g

298g

297g

12h

297g

298g

297g

24h

297g

298g

297g

       According to the above analysis, the moisture of the filler can be dried in an oven for 80 oC at 12h, and the residual water in fillers is almost negligible.

       Thanks for the referee’s kind suggestion. The added formulations of EVA composites are shown in Table 2.

Table 2. Formulations of EVA composites

Compositions/phr

EVA/phr

Mica/phr

APP/phr

ATH/phr

EVAS1

100

100

0

0

EVAS2

100

0

100

0

EVAS3

100

0

-

100

Whether the prepared composite material can form ceramic body is the first key performance in the paper. According to the suggestion, Firstly, the composite EVAS1, EVAS2 and EVAS3 are prepared and then the composites are sintered at 1000 °C for 60min to test whether the ceramic can be formed at high temperatures. The photographs of the residue after sintered at 1000 ° C for 60min are shown in below in Fig. 3, Fig. 4 and Fig. 5. As for the composite EVAS1 containing only EVA and mica, the surface of the sintered product had obvious cracks and also adhered to the firebrick, which could not meet the practical applications. As for the composite EVAS2 containing only EVA and APP, only a small amount of carbon residue was shown in the sintered product, which could not form a complete ceramic body. Regarding the composite EVAS3 containing only EVA and ATH, a large amount of white powder is left in the firebrick and the residue has not any strength. In conclusion, the three experimental formulas could not transform into the ceramic at high temperatures. Therefore, in order to consider the importance of forming the ceramic body at high temperature, so we have not continued to study these three groups of formulas.

Fig. 3. The photograph of the sintered samples EVAS1 containing only EVA and mica

Fig. 4. The photograph of the sintered sample EVAS2 containing only EVA and APP

Fig. 5. The photograph of the sintered sample EVAS3 containing only EVA and ATH

4.     Referring to micro-cone (MCC) tests, most scientific literature agree in considering MCC tests unsuitable for fully describing the behaviour of intumescent or charring-containing flame retardant systems. In addition, the collected data reported in Table 4 are non-sense since the experimental error, calculated as standard deviation of the performed replica, is not reported.

Response: Thanks for the referee’s kind suggestion. We are very sorry for the    missing the experimental error in the MCC tests in our manuscript and we have added an error bar to the MCC data in Table 4 and marked in red in Line 196, Page 6. The cone calorimeter technique is the most advanced method for assessing materials’ reaction to fire on a small scale at present, so we have added the cone calorimetry test in the manuscript and marked in blue in the part of “Combustion performance of ceramifiable flame-retardant EVA composites” in Line 189, Page 6.

5.     Cone calorimetry would be a more suitable and sophisticated technique and I suggest to carry out these tests under a 50 kW/m2 heat flow.

Response: We are very grateful to this meaningful advice for the manuscript. The first reviewer pointed that “It could be also useful to perform forced-combustion (i.e. cone calorimetry) tests, starting with a heat flux of 35 kW/m2”. Hence, we have done related tests and added the related results in our manuscript, which is marked in blue in the discussion in the part “3.3 Combustion performance of ceramifiable flame-retardant EVA composites in Line 189, Page 6. In addition, according to the suggestion, we also have done the cone calorimetry test a heat flux of 50 kW/m2 and added the related results in the “Supporting Information”. The detailed discussion was shown in below.

Figure S1 and Table S1 present the results of cone calorimeter experiments of EVA composites at a flux of 50 kW m-2. It could be seen that the PHRR value of EVA1 was 217.4 kW m-2 and the PHRR value of EVA2 was 168.1 kW m-2. It could be seen that the PHRR value of EVA5 with APP and ATH was 178.9 kW m-2, which is between that of EVA1 only with APP (217.4 kW m-2) and that of EVA2 only with ATH (170.8 kW m-2).     

Figure S1(b) presents the total heat release (THR) curves of EVA composites at a flux of 50 kW m-2. Compared with EVA2 (73.2 MJ m-2), the THR value of EVA1 (63.7 MJ m-2) only with APP was reduced. Regarding the EVA5 sample, the THR value of EVA (58.3 MJ m-2) was lowest among all the composites. Hence, the THR values also demonstrates that EVA5 composites had better fire safety than the other composites presented in Table S1.

The smoke production rate (SPR) curves of EVA composites are illustrated in Figure S1(c). It could be observed that the peak SPR values of EVA1 (2.2×10-2 m2 s-1) and EVA5 (2.4×10-2 m2 s-1) composites containing APP were higher than that the EVA2 only with ATH (1.2×10-2 m2 s-1) in the combustion process. The results could be explained by the decomposition of APP at low temperature to form some smoke particles [7].

The mass loss curves of all the EVA composites in the CCT is shown in Figure S1(d). Sample EVA5 lost its mass, which speed was slower than that of EVA1 and EVA2. Compared with that of EVA1 (59.3%), the char residue of the EVA5 (61.3%) only with APP improved. In addition, the mass loss rate of sample EVA5 was slower than EVA1 and EVA2. The char residue of sample EVA2 was the lowest among all the samples, which was attributed to the absence of APP acting as charring agent in the system.

Based on the above analysis, the sample EVA5 with APP and ATH had better flame retardancy than EVA1 and EVA2. This result based on heat flux of 50 kW/m2 was different from the result obtained at 35 kW/m2, which may be because the prepared composite was more capable of forming dense carbon residue at higher temperatures,

Figure S1. HRR (a), THR (b), SPR (c), and ML (d) curves of neat EVA and EVA

composites at a flux of 50 kW m-2.

Table S1. CC data of EVA composites at a flux of 50 kW m-2

Samples

PHRR   (kW m-2)

THR   (MJ m-2)

TTI   (s)

Residue   (wt%)

Peak   SPR (1 ×10-2 m2 s-1)

EVA1

217.4

63.7

   64

59.3

2.2

EVA2

170.8

73.2

   66

56.8

1.2

EVA5

178.9

58.3

   73

61.3

2.4

Figure S2 is the photographs of char residues for the EVA composites after cone calorimeter test. From Figure S1 (b), a grayish white color of the residue was formed and big holes or cracks could be also observed. In Figure S2(a), compact and continuous char layers were formed for EVA1 and EVA5 composite.

The scanning micrographs of char residues for the EVA1 and EVA5 composites after cone calorimeter test at a flux of 50 kW m-2 are shown in Figure S3. The composite EVA1 residue showed a continuous structure with a number of holes, while the carbonaceous residues of EVA5 with APP and ATH became more compact. Hence, the sample EAV5 had the densest char residue among all the samples, which could explain the lowest HRR, THR, and a series of data tested by cone calorimeter.

Figure S2. The digital photographs of the residues at a flux of 50 kW m-2.

 (a), EVA1; (b), EVA2; (c), EVA5;

Figure S3. Surface morphologies of the residues after cone calorimetry. (a,c), EVA5; (b,d), EVA1;

    [7] Xilei Chen, Yufeng Jiang, Chuanmei Jiao. Smoke suppression properties of ferrite yellow          on flame retardant thermoplastic polyurethane based on ammonium polyphosphate[J].          Journal of Hazardous Materials.2014(266) 114-121.

6.     Furthermore, as the Authors cited, EVA copolymers found a wide application for cables and wires. To this purpose, UL94 classification or rating is not suitable. In fact, for this application, the right tests are the Vertical Flame Test that follows the BSEN 60331-1-2 standard, which evaluates the flame retardant properties on a single cable length, or the UL VW-1 classification, also known as UL 1581, which is the American standard for cables and wires. My suggestion is to perform one of these tests, or at least any other standard test reported as suitable for this application.

Response: Thanks for the referee’s good comment. The test characterization of UL-1581 is indeed a good indicator of the performance of the cable, and the reviewer provide a good guide for our subsequent research. We are very sorry for that we are not capable of doing this test because our school and some cooperative research departments do not have instruments to test the flame retardant on a single cable length. The next step in our research is expected to cooperate with cable manufacturers to make the composite material into a cable to test the flame retardant to meet the practical application.

At last, thank you again for your precious time and efforts for the publication of this paper.

Reviewer 2 Report

The subject of the paper is not completely original. The thermal properties and fire reaction of APP/ATH/EVA composites were the subject of several studies. Nevertheless the paper is well structured. The experimental description and the result analyses are clear.

Author Response

(The authors gave the same response as above.)

Reviewer 3 Report

The submitted paper is focused on a topic that has been widely proposed in the scientific literature. The combination of ammonium polyphosphate and aluminium hydroxide has been already proven to be an efficient flame retardant for many polymers and in particular for EVA copolymers. The novelty of the paper is the use of fluorophlogopite mica, but its specific role is not sufficiently highlighted and discussed in detail.

The term “ceramifiable” is not commonly used in the field of polymer flame retardancy. May be the Authors wanted to refer to the “charring effect” of the proposed flame-retardants. This aspect should be clarified.

As far as the sample preparation is concerned, two aspects should be clarified: i) the residual water content has been missed for the fillers after drying, but is an important information for further compounding; ii) the range of sample compositions should be completed. For instance, EVA1 contains 100 phr of EVA and 100 phr of both mica and APP, and EVA2 contains 100 phr of EVA and 100 phr of both mica and ATH. What is the behaviour of the sample containing only EVA and mica? Or only EVA and ATH, or EVA and APP? Testing these formulations would allow understanding the bearing of each component on fire retardancy and/or the occurrence, if any, of synergistic effects among them.

Referring to micro-cone (MCC) tests, most scientific literature agree in considering MCC tests unsuitable for fully describing the behaviour of intumescent or charring-containing flame retardant systems. In addition, the collected data reported in Table 4 are non-sense since the experimental error, calculated as standard deviation of the performed replica, is not reported.

Cone calorimetry would be a more suitable and sophisticated technique and I suggest to carry out these tests under a 50 kW/m2 heat flow.

Furthermore, as the Authors cited, EVA copolymers found a wide application for cables and wires. To this purpose, UL94 classification or rating is not suitable. In fact, for this application, the right tests are the Vertical Flame Test that follows the BSEN 60331-1-2 standard, which evaluates the flame retardant properties on a single cable length, or the UL VW-1 classification, also known as UL 1581, which is the American standard for cables and wires. My suggestion is to perform one of these tests, or at least any other standard test reported as suitable for this application.

Author Response

(The authors gave the same response as above.)

Round 2

Reviewer 1 Report

The manuscript has been revised according to the Reviewers' comments and suggestions; therefore, now it seems suitable for publication in Polymers.

Reviewer 3 Report

Dear Authors,

I am satisfied for your response and recommend you paper as it is now for publication.